# Scaling Agentic Verifier for Competitive Coding

**Zeyao Ma** [1 2 3]  **Jing Zhang** [1 4]  **Xiaokang Zhang** [1]  **Jiaxi Yang** [3]  **Zongmeng Zhang** [3]  **Jiajun Zhang** [3]
**Yuheng Jing** [3]  **Lei Zhang** [3]  **Hao Zheng** [3]  **Wenting Zhao** [3]  **Junyang Lin** [3]  **Binyuan Hui** [3]

## Abstract

Large language models (LLMs) have demonstrated strong coding capabilities but still struggle to solve competitive programming problems correctly in a single attempt. Execution-based re-ranking offers a promising test-time scaling strategy, yet existing methods are constrained by either difficult test case generation or inefficient random input sampling. To address this limitation, we propose **Agentic Verifier**, an execution-based agent that actively reasons about program behaviors and searches for highly discriminative test inputs that expose behavioral discrepancies among candidate solutions. Through multi-turn interaction with code execution environments, the verifier iteratively refines the candidate input generator and produces targeted counterexamples rather than blindly sampling inputs. We train the verifier to acquire this discriminative input generation capability via a scalable pipeline combining large-scale data synthesis, rejection fine-tuning, and agentic reinforcement learning. Extensive experiments across five competitive programming benchmarks demonstrate consistent improvements over strong execution-based baselines, achieving up to **+10–15%** absolute gains in Best@$k$ accuracy. Further analysis reveals clear test-time scaling behavior and highlights the verifier's broader potential beyond reranking.

## 1. Introduction

Large language models (LLMs) (Achiam et al., 2023) have achieved remarkable progress in a wide range of coding

tasks, including program synthesis (Chen, 2021), completion (Izadi et al., 2024), translation (Pan et al., 2023), and editing (Cassano et al., 2024). Among these tasks, competitive programming (Li et al., 2022) remains particularly challenging, as it requires deep algorithmic understanding, long-horizon reasoning, and precise handling of edge cases. Although recent advances in test-time scaling (e.g., long chain of thought reasoning (Wei et al., 2022; Jaech et al., 2024; Guo et al., 2025)) have substantially improved performance, producing a correct solution in a single attempt is still difficult for even the strongest LLMs (Li et al., 2022).

A common strategy to further boost accuracy is to sample multiple candidate solutions and apply a verifier to select the best one (Cobbe et al., 2021; Wang et al., 2023; Lightman et al., 2024). Execution-based verification (i.e., model-generated test cases) (Chen et al., 2023) has proven especially effective in many code generation settings. However, existing verification approaches are ill-suited for competitive programming. Generating complete test cases with both inputs and ground-truth outputs is often as difficult as solving the original problem itself (Ma et al., 2025), making such methods computationally expensive and unreliable. To mitigate the high cost and difficulty of generating full test cases, recent work has explored input-only execution (Li et al., 2022; Shi et al., 2022): generating valid test inputs, executing candidate solutions on them, and selecting solutions via output agreement or majority voting.

Despite its simplicity, input-only verification typically relies on randomly generated test inputs. In practice, we find this approach to be both inefficient and suboptimal. The space of valid inputs in competitive programming is extremely large, while only a small fraction of inputs are truly discriminative (i.e., capable of exposing behavioral differences between correct and incorrect solutions). As a result, random input generators rarely sample the critical corner cases needed to effectively distinguish correct solutions from incorrect ones, even when scaled to hundreds of executions. Our preliminary experiments across multiple competitive programming benchmarks confirm a substantial performance gap between random inputs and carefully designed ground-truth test cases. This observation motivates a fundamental question: *rather than relying on blind random sampling, can we train a verifier to actively search for highly distin-*

[1]School of Information, Renmin University of China, Beijing, China [2]Key Laboratory of Data Engineering and Knowledge Engineering, Beijing, China [3]Qwen Team, Alibaba Group, Beijing, China [4]Engineering Research Center of Database and Business Intelligence, Beijing, China. Correspondence to: Jing Zhang <zhang-jing@ruc.edu.cn>.

*Proceedings of the 43rd International Conference on Machine Learning*, Seoul, South Korea. PMLR 306, 2026. Copyright 2026 by the author(s).

*guishable test inputs that maximize behavioral divergence among candidate solutions?*

In this work, we introduce **Agentic Verifier**, an LLM-based agent designed to intelligently generate valid test inputs that effectively separate candidate programs. Given a problem and a pair of candidate solutions, the verifier interacts with a code execution environment in multiple turns, reasoning about program behaviors and iteratively refining input generators to discover counterexamples. To train this verifier, we develop a multi-stage pipeline that includes large-scale data synthesis, rejection fine-tuning on successful interaction trajectories, and agentic reinforcement learning to further improve discriminative power.

We evaluate our approach across two policy models (i.e., models that generate candidate code solutions) and five competitive programming benchmarks spanning diverse difficulty levels and sources. Our agentic verifier consistently outperforms existing verification methods, including random input generation and model-generated test cases. Furthermore, our method benefits continuously from increased test-time computation, scaling effectively with both the number of candidate solutions and the number of generated inputs. Beyond improving best-of-$N$ selection, we also analyze the limitations of using fixed benchmark test suites as correctness oracles. In competitive programming, a solution is typically deemed correct if it passes all provided test cases. However, these test suites cover only a limited portion of the valid input space and may miss critical corner cases, leading to false-positive solutions that pass the tests but are incorrect on other valid inputs. Our agentic verifier can actively expose such failures by discovering targeted counterexamples that reveal behavioral discrepancies, suggesting its broader utility as a complementary correctness-checking tool rather than merely a reranking component.

**Contributions**   In summary, our contributions are: (a) We empirically demonstrate that random input generation is highly inefficient for input-only execution-based verification in competitive programming, revealing a large performance gap compared to discriminative test inputs. (b) We propose an agentic verifier that actively generates highly discriminative test inputs through multi-turn interaction with code execution environments. (c) We introduce a scalable training pipeline combining data synthesis, rejection fine-tuning, and agentic reinforcement learning. (d) Extensive experiments show substantial improvements across multiple competitive programming benchmarks, with clear test-time scaling behavior and robustness to problem difficulty.

**Conflict of Interest Disclosure**   The authors H.Z., W.Z., J.L., and B.H. are employed by Alibaba Group, and Z.M., J.Y., Z.Z., J.Z., Y.J., and L.Z. interned at Alibaba Group, which contributes to the development of Qwen3 models, which were among the ones evaluated in this paper.

## 2. Observation

In this section, we examine how the source of test inputs influences best-of-$N$ performance under input-only execution-based voting. Specifically, we compare randomly generated inputs with the ground-truth inputs provided by the original benchmark suites. By evaluating across three competitive coding benchmarks of varying difficulty, we aim to understand whether scaling random inputs is sufficient for reliable solution selection. In addition, we include results obtained using our trained agentic verifier to provide further context on the potential of learned, discriminative input generation.

### 2.1. Execution-based Voting Method

We first introduce our best-of-$N$ voting method based on test input execution. Given a problem $Q$, a policy model samples $N$ candidate solutions $C = \{C_1, \ldots, C_N\}$. We also construct a set of $M$ test inputs (without ground-truth outputs) $X = \{x_1, \ldots, x_M\}$. Each candidate solution is executed on each test input, producing an output

$$o_{i,j} = \text{Exec}(C_i, x_j),$$

where $o_{i,j} = \bot$ indicates execution failure such as runtime error or timeout.

Our voting strategy relies on output agreement across candidate solutions. For each test input $x_j$, solutions that produce identical outputs are grouped into the same cluster. The agreement score of candidate $C_i$ on input $x_j$ is defined as the size of its output cluster:

$$v_{i,j} = \left| \{k \in \{1, \ldots, N\} : o_{k,j} = o_{i,j}\} \right|.$$

Candidates that achieve the maximum agreement on a given input receive one vote (ties allowed). The overall score of each candidate aggregates its wins across all test inputs:

$$s_i = \sum_{j=1}^{M} \mathbf{1} \left[ v_{i,j} = \max_{t \in \{1,\ldots,N\}} v_{t,j} \right].$$

The final selected solution is the candidate with the highest total score.

The key intuition behind this voting strategy is that, on a valid and discriminative test input, all correct solutions produce the same output and thus form a consistent cluster, whereas incorrect solutions tend to produce diverse erroneous outputs that scatter across smaller clusters. By aggregating votes over many discriminative inputs, the correct cluster tends to accumulate the most votes, enabling reliable selection even in the presence of occasional invalid or non-discriminative inputs.

We instantiate the test input set $X$ using two sources: randomly generated test inputs and the ground-truth test inputs from the benchmark suites. We vary the number of test inputs $M$ and report the resulting best-of-$N$ performance.

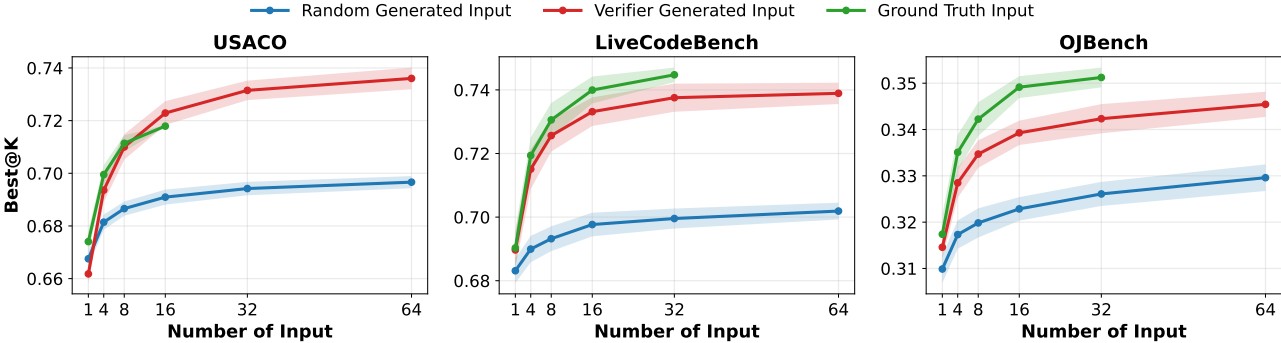

*Figure 1.* Best-of-$N$ performance under input-only execution-based voting as the number of test inputs increases. We compare randomly generated inputs, ground-truth test inputs from the benchmark suites, and inputs generated by our trained agentic verifier. Ground-truth inputs consistently outperform random inputs by a large margin, indicating the inefficiency of naive random input scaling. Verifier-generated inputs significantly improve over random generation and even surpass ground-truth test cases on USACO, demonstrating stronger discriminative power.

### 2.2. Preliminary Experiments

**Settings**   We conduct preliminary experiments on three competitive programming benchmarks spanning diverse difficulty levels. USACO (Shi et al., 2024) consists of problems from the USA Computing Olympiad. LiveCodeBench (version 6) (Jain et al., 2025) contains recent problems collected from LeetCode and AtCoder starting from 2025-02. OJBench (Wang et al., 2025b) includes challenging problems from multiple sources, including NOI and ICPC.

We use Qwen3-30B-A3B-Thinking-2507 (Yang et al., 2025) as the policy model to sample $64$ candidate solutions for each problem. For random input generation, we prompt the model with the template in Figure 7 to produce 8 input generators, each sampling 8 test inputs (64 in total). Ground-truth test inputs are taken directly from the original benchmark suites. All methods apply the execution-based voting strategy described in Section 2.1.

**Result and Discussion**   Figure 1 presents best-of-$N$ performance as we vary the number of test inputs for different input sources. Across all three benchmarks, ground-truth test inputs consistently achieve substantially higher performance than randomly generated inputs, even when using only a small number of test cases. In contrast, scaling random inputs yields only modest improvements and fails to close the performance gap. For example, on USACO, using 16 ground-truth inputs already outperforms voting with 64 randomly generated inputs by a large margin. Similar trends are observed on LiveCodeBench and OJBench, indicating that naive random input scaling is highly inefficient for verification in competitive programming.

We further include results obtained using our trained agentic verifier. Verifier-generated inputs significantly outperform random inputs across all benchmarks and, on USACO, even surpass the performance achieved using ground-

truth test inputs. On LiveCodeBench and OJBench, while slightly below ground-truth inputs, verifier-generated inputs remain substantially more effective than random generation. Overall, these findings demonstrate that the effectiveness of execution-based voting is primarily determined by the discriminative quality of test inputs rather than their sheer quantity, motivating the need to train models that can actively generate targeted and highly discriminative inputs.

## 3. Training Agentic Verifier

The observations in Section 2 indicate that the effectiveness of input-only execution-based voting is primarily constrained by the quality and discriminative power of test inputs. This motivates the development of a learned verifier that can actively search for valid and highly distinguishable test inputs. In this section, we present the training framework for our agentic verifier: we formalize the multi-turn input generation task (§3.1), describe the large scale data synthesis process (§3.2), and introduce the training pipeline, including rejection fine-tuning (§3.3) and agentic reinforcement learning (§3.4).

### 3.1. Task Definition

We formulate distinguishable test input generation as a multi-turn, execution-based interaction task. Each task instance consists of a competitive programming problem $Q$ and a pair of candidate solutions $(C_a, C_b)$. The objective of the agentic verifier is to produce an input generator program $G$ that, when executed, samples a valid test input $x = G()$ capable of distinguishing the two solutions.

The verifier interacts with a code execution environment through tool calls across multiple turns (see Figure 9 for tool definition). During the interaction, the model may execute candidate solutions on proposed inputs, inspect exe-

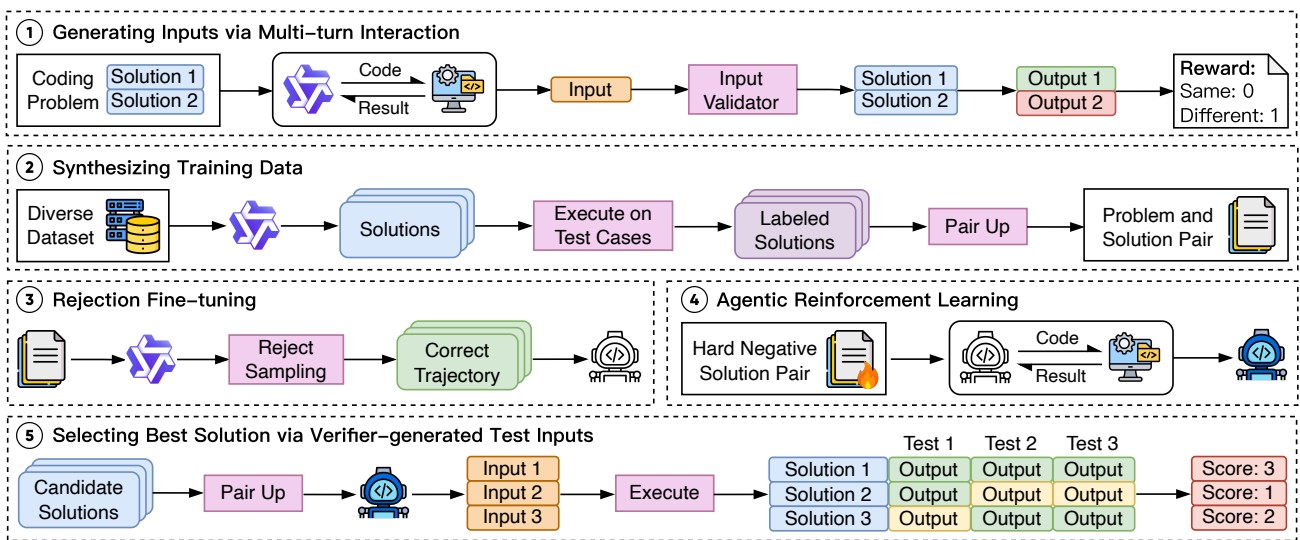

*Figure 2.* Overview of the agentic verifier training and inference pipeline. The verifier is trained on a multi-turn input generation task through large-scale data synthesis, rejection fine-tuning on successful interaction trajectories, and agentic reinforcement learning with hard negative solution pairs. At test time, the trained verifier generates discriminative test inputs for execution-based voting to select the best candidate solution.

cution feedback and intermediate results, and verify whether generated inputs satisfy the problem constraints. These interactions are primarily used for reasoning and exploration, enabling the verifier to understand program behaviors and refine its hypotheses about potential failure cases.

After sufficient interaction, the verifier outputs a final input generator program. Executing this generator produces a test input, which is then evaluated on both solutions. A generated input is considered *valid* if it satisfies the input constraints specified by the problem $Q$. Among valid inputs, an input is deemed *distinguishing* if $\mathrm{Exec}(C_a, x) \neq \mathrm{Exec}(C_b, x)$. The verifier succeeds on a task instance if the produced generator can yield at least one valid distinguishing input. This task directly optimizes the verifier for generating highly discriminative test inputs through execution-based multi-turn interaction.

We adopt the pairwise formulation because distinguishing two solutions is a simpler binary comparison problem than jointly distinguishing among multiple candidates. At inference time, this connects to multi-candidate selection through the voting mechanism in Section 2.1: we sample candidate pairs, generate one discriminative input per pair, and then execute all $N$ candidates on all generated inputs, aggregating the results into a global ranking via output agreement.

### 3.2. Training Data Synthesis

To support training on the multi-turn input generation task, we construct a large-scale dataset of competitive programming problems with labeled candidate solutions and validated execution environments. We collect problems from

multiple sources, including open-source competitive programming datasets with reliable test cases and problems crawled from online judge platforms. For online judge problems, we obtain labeled solutions through web crawling, which include both purportedly correct and incorrect submissions, and additionally incorporate a small set of human-authored correct reference solutions. These diverse solutions provide a foundation for synthesizing training instances with rich execution signals.

To prevent data leakage, we perform explicit deduplication between the training data and all evaluation benchmarks involved in this paper (Section 4). Specifically, we compute pairwise problem similarity using Qwen3-Embedding-8B (Zhang et al., 2025c), which achieves state-of-the-art performance on MTEB (as of June 5, 2025) (Muennighoff et al., 2023), and remove all training problems whose similarity to any evaluation problem exceeds $0.7$.

For problems without accessible ground-truth test cases, we synthesize high-quality test case sets. We first prompt an LLM to generate candidate test inputs and execute them on the reference correct solutions to obtain corresponding outputs. To ensure reliability, we apply a consensus-based majority voting mechanism: for each generated test input, we evaluate it across multiple reference solutions and retain only those inputs for which a large fraction of reference solutions produce consistent outputs, using a high agreement threshold. Inputs that fail this consistency check are discarded, as they often indicate special-judge problems or mislabeled reference solutions. We further refine the test case sets by selecting inputs that maximally reject incorrect solutions while preserving agreement among correct ones,

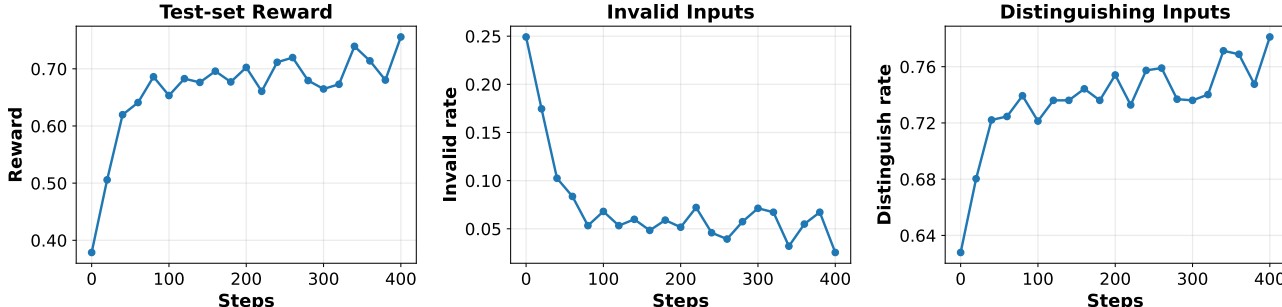

*Figure 3.* Training dynamics of agentic reinforcement learning on a held-out test set. We report the reward over training steps (left), along with the rates of invalid inputs (middle) and distinguishing inputs (right), computed from test-set rollouts. Reinforcement learning steadily improves discriminative input generation while reducing invalid outputs, indicating stable and effective training.

resulting in high-quality and discriminative test suites with an average of 60 test cases per problem.

Using these constructed test cases, we generate multiple candidate solutions with Qwen3-30B-A3B-Thinking-2507 and label them according to their execution results. We subsequently form training pairs $(C_a, C_b)$ such that at least one test input produces different outputs between the two solutions, ensuring the existence of a valid distinguishing input for each instance. For each problem, we construct LLM-generated input validators to verify input constraints (see Figure 8 for the prompt used to generate input validators), retaining only those consistent with all high-confidence synthesized test cases, and determine distinguishability via exact output matching.

### 3.3. Rejection Fine-tuning

As the input generation task requires multi-turn reasoning and interaction with execution environments, we first bootstrap the agentic verifier through rejection fine-tuning on successful interaction trajectories. We implement a secure code execution sandbox supporting both Python and C++ to enable large-scale trajectory collection. Using Qwen3-235B-A22B-Thinking-2507 as a strong teacher model, we sample diverse multi-turn interaction traces for each task instance $(Q, C_a, C_b)$ constructed in Section 3.2. Each trace consists of a sequence of tool calls, intermediate execution feedback, and a final input generator program.

We retain only trajectories that successfully produce valid and distinguishing inputs, discarding failed attempts. From these filtered trajectories, we collect approximately 60K high-quality examples and perform supervised fine-tuning on a continued-pretrained Qwen3-30B-A3B base model. During training, we mask the initial prompts and execution feedback turns, and optimize the model to predict the agent's actions and final generator programs, enabling the verifier to learn effective interaction patterns and generate discriminative input generators.

### 3.4. Agentic Reinforcement Learning

Following rejection fine-tuning, we further improve the verifier using agentic reinforcement learning to enhance its ability to generate highly discriminative input generator programs. We formulate the distinguishable input generation task as a trajectory-level optimization problem, where each rollout consists of a sequence of tool interactions and a final generated input generator $G$, which produces a concrete test input $x = G()$.

We define a sparse, outcome-based reward function based on the validity and discriminative power of the generated input:

$$r(x) = \begin{cases} -1, & \text{if } x \text{ is invalid or } G \text{ is empty,} \\ 0, & \text{if } \text{Exec}(C_a, x) = \text{Exec}(C_b, x), \\ 1, & \text{if } \text{Exec}(C_a, x) \neq \text{Exec}(C_b, x). \end{cases}$$

This reward directly encourages the verifier to discover valid inputs that expose behavioral differences between candidate solutions.

We adopt GRPO (Shao et al., 2024) as the optimization algorithm for multi-turn reinforcement learning. To construct effective training queries, we prioritize difficult task instances based on their pass rates observed during rejection sampling, as well as hard negative solution pairs that admit only a small number of distinguishing inputs. This curriculum focuses on challenging cases where naive generation rarely produces distinguishing inputs, encouraging the verifier to learn more effective and targeted input generation behaviors. We train the verifier for 400 steps on 10K selected queries, reserving 500 queries as a held-out evaluation set for monitoring training progress. As illustrated in Figure 3, reinforcement learning steadily increases the verifier's ability to generate distinguishing inputs while significantly reducing the rate of invalid inputs, demonstrating improved generation quality and discriminative capability.

*Table 1.* Main results of test-time scaling across competitive coding benchmarks. We report Best@$k$ performance ($k \in \{8, 64\}$) for two policy models, comparing vanilla sampling, reward-model-based selection, execution-based baselines, and the proposed agentic verifier. The agentic verifier consistently achieves the strongest performance across all settings.

| Method / Dataset | USACO | | LiveCodeBench | | OJBench | | ICPC-Eval | | CodeForces | |
|---|---|---|---|---|---|---|---|---|---|---|
| | Best@8 | Best@64 | Best@8 | Best@64 | Best@8 | Best@64 | Best@8 | Best@64 | Best@8 | Best@64 |
| Qwen3-30B-A3B-Thinking-2507 | | | | | | | | | | |
| Vanilla | 62.8 | 62.8 | 66.2 | 66.2 | 28.9 | 28.9 | 16.6 | 16.6 | 33.2 | 33.2 |
| Grading RM | 63.0 +0.2 | 61.9 -0.9 | 67.1 +0.9 | 68.0 +1.8 | 28.7 -0.2 | 29.3 +0.4 | 18.1 +1.5 | 13.6 -3.0 | 33.0 -0.2 | 32.8 -0.4 |
| MBR-Exec (hard) | 64.4 +1.6 | 68.5 +5.7 | 68.5 +2.3 | 69.2 +3.0 | 29.6 +0.7 | 32.4 +3.5 | 17.7 +1.1 | 19.8 +3.2 | 33.4 +0.2 | 33.7 +0.5 |
| MBR-Exec (soft) | 67.9 +5.1 | 69.6 +6.8 | 70.0 +3.8 | 70.0 +3.8 | 31.4 +2.5 | 31.8 +2.9 | 20.5 +3.9 | 20.5 +3.9 | 38.3 +5.1 | 41.8 +8.6 |
| CodeT | 68.5 +5.7 | 69.8 +7.0 | 72.7 +6.5 | 70.5 +4.3 | 31.6 +2.7 | 32.1 +3.2 | 21.6 +5.0 | 21.3 +4.7 | 38.6 +5.4 | 40.1 +6.9 |
| CodeRM | 68.7 +5.9 | 68.4 +5.6 | 71.2 +5.0 | 71.4 +5.2 | 32.0 +3.1 | 31.3 +2.4 | 21.3 +4.7 | 18.5 +1.9 | 38.9 +5.7 | 42.3 +9.1 |
| Random Generator | 69.7 +6.9 | 70.2 +7.4 | 70.7 +4.5 | 70.4 +4.2 | 33.4 +4.5 | 34.1 +5.2 | 22.4 +5.8 | 22.2 +5.6 | 39.5 +6.3 | 40.1 +6.9 |
| Agentic Verifier | 73.1 +10.3 | 74.5 +11.7 | 73.6 +7.4 | 73.7 +7.5 | 33.5 +4.6 | 34.9 +6.0 | 23.1 +6.5 | 25.8 +9.2 | 43.0 +9.8 | 46.4 +13.2 |
| Qwen3-235B-A22B-Thinking-2507 | | | | | | | | | | |
| Vanilla | 75.4 | 75.4 | 74.7 | 74.7 | 37.1 | 37.1 | 23.6 | 23.6 | 39.7 | 39.7 |
| Grading RM | 75.8 +0.4 | 73.6 -1.8 | 75.9 +1.2 | 74.9 +0.2 | 37.6 +0.5 | 34.9 -2.2 | 23.5 -0.1 | 22.0 -1.6 | 39.7 +0.0 | 34.4 -5.3 |
| MBR-Exec (hard) | 75.8 +0.4 | 79.2 +3.8 | 77.9 +3.2 | 77.7 +3.0 | 37.7 +0.6 | 39.6 +2.5 | 23.9 +0.3 | 27.4 +3.8 | 40.0 +0.3 | 42.2 +2.5 |
| MBR-Exec (soft) | 79.9 +4.5 | 80.3 +4.9 | 78.5 +3.8 | 78.5 +3.8 | 40.5 +3.4 | 41.4 +4.3 | 26.2 +2.6 | 26.4 +2.8 | 43.8 +4.1 | 45.4 +5.7 |
| CodeT | 79.0 +3.6 | 80.4 +5.0 | 78.8 +4.1 | 79.2 +4.5 | 41.0 +3.9 | 40.6 +3.5 | 26.9 +3.3 | 27.4 +3.8 | 44.6 +4.9 | 46.9 +7.2 |
| CodeRM | 78.7 +3.3 | 77.5 +2.1 | 79.3 +4.6 | 80.3 +5.6 | 40.4 +3.3 | 39.5 +2.4 | 26.7 +3.1 | 27.1 +3.5 | 45.9 +6.2 | 45.1 +5.4 |
| Random Generator | 81.8 +6.4 | 82.2 +6.8 | 78.8 +4.1 | 78.9 +4.2 | 43.8 +6.7 | 44.2 +7.1 | 28.7 +5.1 | 29.7 +6.1 | 46.7 +7.0 | 45.4 +5.7 |
| Agentic Verifier | 83.0 +7.6 | 84.1 +8.7 | 81.6 +6.9 | 82.9 +8.2 | 45.9 +8.8 | 48.6 +11.5 | 29.9 +6.3 | 31.2 +7.6 | 49.9 +10.2 | 50.9 +11.2 |

# 4. Experiments

## 4.1. Experimental Setup

**Benchmarks** We conduct extensive evaluations on four established competitive programming benchmarks covering a wide range of problem difficulties, including US-ACO (Shi et al., 2024), LiveCodeBench (Jain et al., 2025), OJBench (Wang et al., 2025b), and ICPC-Eval (Xu et al., 2026). USACO contains 307 problems from the USA Computing Olympiad. LiveCodeBench consists of recent problems collected from LeetCode, AtCoder, and CodeForces; to avoid potential data contamination, we select 175 problems starting from February 2025. OJBench includes 232 challenging problems sourced from NOI and ICPC. ICPC-Eval contains 118 problems from ICPC contests. Overall, USACO and LiveCodeBench are relatively easier, while OJBench and ICPC-Eval represent harder settings.

In addition to existing benchmarks, we introduce a new benchmark called **CodeForces** to further assess generalization. While current benchmarks primarily draw from well-known contests such as USACO, NOI, and ICPC, they lack contests with high-quality test cases sourced from CodeForces[1], which is a well-known competitive coding platform. We therefore curate 8 CodeForces contests comprising 64 problems starting from December 2024, and construct their test cases using the synthesis procedure described in Section 3.2.

---
[1] https://codeforces.com/

**Baselines** We evaluate our method on two policy models, Qwen3-30B-A3B-Thinking-2507 (Yang et al., 2025) and Qwen3-235B-A22B-Thinking-2507, and compare against a diverse set of baselines. The Vanilla baseline randomly selects one candidate solution. Grading RM assigns a scalar score to each solution and selects the highest-scoring one. MBR-Exec (hard and soft variants) (Shi et al., 2022) reranks solutions using model-generated test inputs. CodeT (Chen et al., 2023) and CodeRM (Ma et al., 2025) leverage model-generated test cases with both inputs and outputs for execution-based reranking. Random Generator (Li et al., 2022) uses model-generated input generator programs to sample test inputs for voting-based selection. Details about baselines are provided in Appendix B.

**Implementation Details and Metrics.** For the grading reward model, we use Skywork-Reward-V2-Llama-3.1-8B (Liu et al., 2025). For all execution-based baselines that require test input or test case generation, we employ Qwen3-30B-A3B-Thinking-2507 with a budget of 512 test inputs or test cases per problem. Our agentic verifier is used to generate test inputs for the proposed method. At inference time, to match this budget, we randomly sample pairs of candidate solutions and apply the verifier to each pair to generate one test input. For the Random Generator and Agentic Verifier, solutions are selected using the execution-based voting strategy described in Section 2.1.

We report Best@$k$ as the primary evaluation metric, defined as the Pass@1 accuracy of the final selected solution from

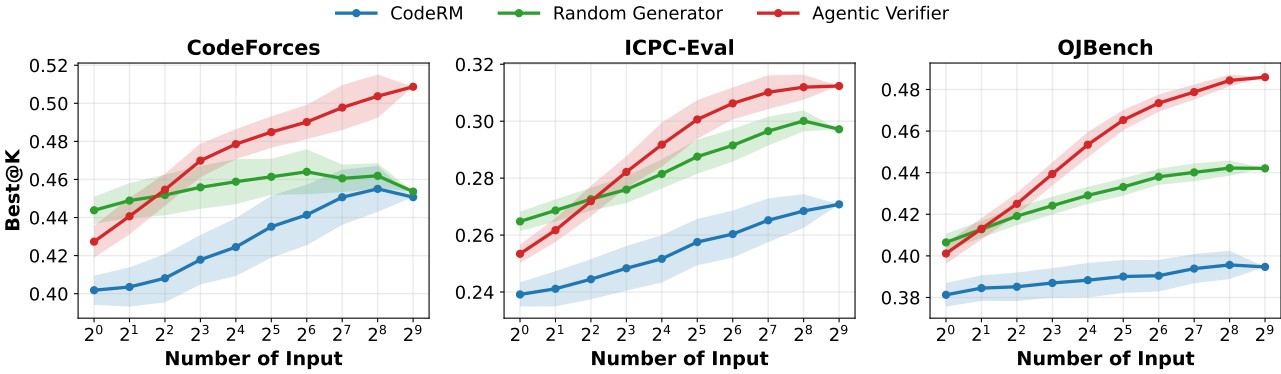

*Figure 4.* Scaling effect of execution-based methods under the Best@64 setting as the number of test inputs increases exponentially. Our agentic verifier exhibits stronger and more stable performance gains compared with representative baselines.

$k$ sampled candidates. We set $k \in \{8, 64\}$. For Best@8, we repeat random sampling of candidate solutions 100 times and report the mean to reduce variance.

### 4.2. Main Results

Table 1 summarizes the main results across all benchmarks and policy models. The proposed agentic verifier consistently achieves the best performance under both Best@8 and Best@64 settings, outperforming all baselines by a clear margin. Among execution-based baselines, the Random Generator approach, which samples inputs via programmatic generators, outperforms methods that directly generate full test cases (e.g., CodeRM) in most cases. However, it still falls significantly short of our proposed agentic verifier, highlighting the benefit of training an agentic verifier for discriminative input generation.

The improvements are particularly pronounced on harder benchmarks such as OJBench and ICPC-Eval, where directly generating complex test cases or test inputs is substantially more challenging. Notably, while several baselines exhibit inconsistent or diminishing returns when increasing the number of candidate solutions from Best@8 to Best@64, our method consistently benefits from scaling, with Best@64 outperforming Best@8 across all settings. This demonstrates that the proposed approach more effectively leverages test-time computation, jointly benefiting from scaling both candidate solutions and discriminative test inputs.

## 5. Analysis

### 5.1. Scaling Effect of Test Input

Figure 4 illustrates how performance scales with the number of test inputs when comparing our method with two representative execution-based baselines. While all approaches benefit from increased test input budgets, their scaling be-

*Table 2.* Scaling performance across different problem difficulty levels on OJBench. The performance gap widens substantially as difficulty increases, highlighting the effectiveness of learned discriminative input generation on challenging problems.

| Method | Easy | Medium | Hard |
|---|---|---|---|
| Qwen3-30B-A3B-Thinking-2507 | | | |
| Vanilla | 93.5 | 52.7 | 7.3 |
| CodeRM | 99.2 +5.7 | 59.0 +6.3 | 14.6 +7.3 |
| Random Generator | 99.5 +6.0 | **82.7** +30.0 | 11.9 +6.0 |
| Agentic Verifier | **100.0** +4.6 | 80.0 +27.3 | **20.7** +13.4 |
| Qwen3-235B-A22B-Thinking-2507 | | | |
| Vanilla | 92.1 | 48.6 | 9.9 |
| CodeRM | 86.8 -5.3 | 53.4 +4.8 | 25.6 +15.7 |
| Random Generator | 99.4 +7.3 | 71.9 +23.3 | 24.7 +14.8 |
| Agentic Verifier | **99.9** +7.8 | **80.3** +31.7 | **44.0** +34.1 |

haviors differ substantially. Our agentic verifier consistently exhibits stronger and more stable scaling trends, achieving larger performance gains as the number of test inputs increases. In contrast, baseline methods show earlier saturation with diminishing marginal improvements at larger budgets. These results indicate that learned, discriminative input generation is substantially more effective at leveraging additional test-time computation.

### 5.2. Scaling Effect Across Problem Difficulty

To analyze how test-time scaling behaves under varying problem difficulty, we conduct experiments on OJBench, which provides a well-balanced distribution of easy, medium, and hard problems with sufficient coverage of challenging cases. Table 2 reports the results across the three difficulty subsets. On easy problems, both Random Generator and our method achieve near-saturated performance, leaving limited room for improvement. As difficulty increases, the performance gap between the proposed agentic

*Table 3.* Comparison between our trained agentic verifier and zero-shot LLMs on best-of-$N$ performance using Qwen3-235B-A22B-Thinking-2507 as the policy model and generating 64 test inputs for re-ranking. The trained verifier consistently outperforms zero-shot LLMs, highlighting the importance of agentic training.

| Method | LiveCodeBench | OJBench |
|---|---|---|
| Vanilla | 74.7 | 37.1 |
| Qwen3-30B-A3B-Thinking | 78.8 +4.1 | 43.3 +6.2 |
| Qwen3-235B-A22B-Thinking | 79.5 +4.8 | 43.2 +6.1 |
| Agentic Verifier (30B-A3B) | **82.4** +7.7 | **47.3** +10.2 |

*Table 4.* Ablation of the training pipeline. All verifiers use Qwen3-30B-A3B as the base model and are evaluated with the Qwen3-235B-A22B policy model under Best@64 with 64 verifier-generated inputs. Gains are computed over the vanilla method.

| Method | LiveCodeBench | OJBench |
|---|---|---|
| Vanilla (no verifier) | 74.7 | 37.1 |
| Zero-shot (Qwen3-30B-A3B) | 78.8 +4.1 | 43.3 +6.2 |
| RFT-only (Qwen3-30B-A3B) | 79.9 +5.2 | 44.6 +7.5 |
| RFT + RL (Qwen3-30B-A3B) | **82.4** +7.7 | **47.3** +10.2 |

verifier and baseline methods widens substantially. In particular, on medium and hard problems, the agentic verifier consistently outperforms baselines by a large margin across both policy models, and the gains are most pronounced on the hard subset. These results indicate that the benefits of learned, discriminative input generation are amplified as problem complexity increases, enabling more effective utilization of test-time scaling in challenging scenarios.

### 5.3. Training vs. Zero-shot

We examine whether strong off-the-shelf LLMs can serve as effective zero-shot verifiers for the input generation task in Section 3.1, or whether task-specific agentic training is necessary. Specifically, we directly apply Qwen3-30B-A3B-Thinking-2507 and Qwen3-235B-A22B-Thinking-2507 as zero-shot agentic verifiers without training. As shown in Table 3, zero-shot LLMs improve best-of-$N$ performance, but the trained agentic verifier consistently achieves substantially higher results on both LiveCodeBench and OJBench. Notably, the 30B trained verifier outperforms the zero-shot 235B model by a large margin, despite being over $7\times$ smaller. These results highlight the importance of task-specific agentic training for effective verification.

### 5.4. Training Pipeline Ablation

To examine the contribution of each training stage, we compare zero-shot, RFT-only, and RFT+RL verifiers. As shown in Table 4, RFT-only improves marginally over the zero-shot baseline, primarily because it still generates a high rate of invalid inputs (Figure 3, middle). Adding RL sub-

stantially improves performance. RL explicitly penalizes invalid outputs (reward $= -1$), which reduces the invalid rate and increases discriminative power. The full RFT+RL pipeline yields improvements of $+7.7$ and $+10.2$ points on LiveCodeBench and OJBench respectively, confirming that both training stages contribute meaningfully.

### 5.5. Discussion of Imperfect Verifiers in Benchmarks

**Benchmark Test Suites as Imperfect Verifiers** In competitive programming benchmarks, correctness is typically determined by a fixed set of test cases: a program is labeled as correct if it passes all tests. From a verification perspective, such test suites act as deterministic verifiers but cover only a limited portion of the valid input space (Stroebl et al., 2026). Consequently, passing the benchmark test suites does not guarantee correctness over all valid inputs, leading to false positives.

This limitation becomes particularly evident for input-only execution-based selection methods that generate new valid inputs and compare candidate outputs without ground-truth labels. Multiple candidates labeled as correct may produce inconsistent outputs on newly generated inputs, causing instability in voting or reranking procedures. In many cases, the root cause lies in the incompleteness of the benchmark verifier rather than the selection mechanism itself.

**Case Study: Revealing False Positives with Verifier-generated Inputs** Figure 5 illustrates a concrete example involving repeated removal of adjacent element pairs to maximize the total absolute-difference score. For the valid input $N = 7$ and $A = [98, 77, 2, 86, 25, 79, 96]$, two candidates marked as correct by the benchmark suite produce different outputs (Candidate 1: 159; Candidate 2: 167). Candidate 1's result is achievable through valid adjacent removals, e.g., $(98, 77) \rightarrow (86, 25) \rightarrow (2, 79)$, yielding 159. In contrast, Candidate 2 optimizes a relaxed formulation using a heap-based prefix structure that fails to enforce the dynamic adjacency constraint. While it matches the benchmark test suites, the verifier-generated input exposes this deviation, revealing Candidate 2 as a false positive.

**Implications Beyond Best-of-$N$ Selection** These observations suggest that benchmark test suites should be treated as imperfect verifiers rather than definitive correctness oracles. The proposed agentic verifier can complement benchmark test suites by actively expanding input coverage and exposing hidden behavioral discrepancies. First, it can act as a test suite augmenter by actively searching for valid, highly discriminative inputs and incorporating them into the evaluation pool, improving coverage and reducing false positives. Second, when a high-confidence reference implementation is available, the verifier can perform direct behavioral comparison by discovering counterexamples that expose discrepancies between candidate solutions and the

reference (Allamanis & Yin, 2025).

Importantly, both use cases avoid the need for generating ground-truth outputs and naturally scale with increased test-time computation budgets. This positions the agentic verifier as a flexible verification component that extends beyond best-of-$N$ sampling and naive execution-based voting, offering a practical mechanism for mitigating benchmark-induced noise in large-scale code evaluation.

## 6. Related Work

**Code Reasoning and Benchmarks**  Early code generation models (Chen, 2021) were primarily evaluated on foundational benchmarks such as HumanEval (Chen, 2021) and MBPP (Austin et al., 2021), which focus on functional correctness over relatively simple function-level tasks. Beyond basic code synthesis, recent research has increasingly targeted more complex programming scenarios (Zhang et al., 2025a), particularly competitive programming (Li et al., 2022). More recently, advanced reasoning language models (Jaech et al., 2024; Guo et al., 2025) leverage test-time computation to substantially improve performance on challenging coding tasks. As the capabilities of LLMs on competitive programming continue to advance, the difficulty of evaluation benchmarks has also been progressively increased. Recent benchmarks (Wang et al., 2025b; Xu et al., 2026; Zheng et al., 2025) increasingly draw problems directly from high-level programming competitions, featuring significantly harder algorithmic challenges.

**Program Reranking Methods**  Best-of-$N$ program reranking methods aim to select the most accurate solution from multiple sampled candidates and can broadly be categorized into execution-based and non-execution-based approaches. Execution-based methods rely on running candidate programs on generated test inputs or test cases to assess correctness or behavioral consistency (Li et al., 2022; Shi et al., 2022; Chen et al., 2023; Huang et al., 2024; Ma et al., 2025). For instance, (Shi et al., 2022) employs minimum Bayes risk decoding based on execution results across multiple test inputs, while (Ma et al., 2025) explores dynamically scaling the number of generated test cases to provide richer execution signals for more reliable selection. In contrast, non-execution-based methods employ learned models to directly score and rank candidate programs without executing them (Inala et al., 2022; Zhang et al., 2023; SHUM et al., 2026). These approaches typically train neural rankers to predict program correctness or error patterns based on code representations and instruction consistency.

**Test Case Generation Methods**  Several works also address test case generation for competitive programming. CodeContests+ (Wang et al., 2025a), HardTests (He et al., 2026), AutoCode (Zhou et al., 2026), and Klear-

CodeTest (Fu et al., 2025) all adopt a *Generator-Validator pipeline*, in which a prompt-based LLM generates candidate test inputs and a separate prompt-based LLM validates whether they satisfy the problem constraints. In contrast, our method formulates discriminative test input generation as a *trainable, multi-turn agentic task*. Through rejection fine-tuning and agentic reinforcement learning with rule-based outcome rewards (RLVR), the capabilities of both generation and validation are internalized into a single trained verifier, without relying on a separate prompt-based validator. Furthermore, our training objective, distinguishing two candidate solutions via execution disagreement, provides a verifiable reward signal that enables scalable RL training without human annotation, which is a fundamental departure from the prompt-engineering-based paradigm of the aforementioned works.

## 7. Limitations and Future Work

**Domain Scope**  The current method is designed and evaluated exclusively on competitive programming. While the underlying task formulation, given a problem and two candidate answers, generating tests to distinguish them, is domain-agnostic, its effectiveness in other code domains remains to be validated. Behavioral equivalence testing is a fundamental need across many code domains (Allamanis & Yin, 2025): for instance, one could train an agentic verifier to generate repo-level unit tests that distinguish between candidate patches, improving verification for automated program repair and code editing. We conduct a preliminary experiment on SWE-bench Verified (Jimenez et al., 2024) (Appendix C), observing modest but positive gains ($+1.6\%$) with a zero-shot equivalence judge. We believe this can be substantially improved through task-specific training, and plan to investigate this direction in future work.

**Verifier as Training Signal**  Our current work focuses on test-time selection. The discriminative inputs and execution feedback generated by the agentic verifier could also serve as supervision signals to improve the underlying policy model through reinforcement learning, potentially creating a virtuous cycle between generation and verification. We leave this exploration to future work.

## 8. Conclusion

We show that naive random input scaling is highly inefficient for execution-based verification in competitive programming. To address this limitation, we propose an agentic verifier that actively generates highly discriminative test inputs, trained via a scalable pipeline combining data synthesis, rejection fine-tuning, and agentic reinforcement learning. Extensive experiments demonstrate consistent and robust test-time scaling improvements across diverse benchmarks.

## Acknowledgements

This work is supported by the National Key Research & Development Plan (2023YFF0725100) and the National Natural Science Foundation of China (92570121, 62322214, U23A20299, U24B20144).

## Impact Statement

This paper presents a method for improving execution-based verification in competitive programming through an agentic verifier that actively generates discriminative test inputs. Our goal is to advance the reliability and scalability of test-time selection and evaluation for large language models in complex competitive coding tasks.

The proposed approach primarily contributes to research on program verification, code generation, and model evaluation, and is intended to improve robustness in automated coding systems. We do not foresee immediate negative societal consequences arising directly from this work. As with other advances in automated software generation and evaluation, potential impacts depend on downstream deployment practices. We believe that improving verification and correctness checking is generally beneficial for the safe and reliable use of AI-assisted programming tools.

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

# A. Case Study: False Positives from Imperfect Benchmark Verifiers

**Extended analysis of the false positive example.** We further analyze the two candidate solutions in Figure 5 to clarify why Candidate Solution 1 is correct while Candidate Solution 2 violates the original problem constraints despite passing the benchmark test suite.

The problem allows repeatedly removing *adjacent* pairs of elements and accumulating the absolute difference of each removed pair. After each operation, the sequence is dynamically shortened and new adjacency relations are formed. Therefore, the core constraint is that each removal must involve two elements that are adjacent at the moment of removal, rather than arbitrary pairs selected from the original array. This dynamic adjacency requirement fundamentally distinguishes the problem from unconstrained pairing or matching formulations.

Candidate Solution 1 correctly models this constraint. For even-length sequences, it computes the optimal score as the difference between the sum of the largest half and the smallest half of the elements after sorting. For odd-length sequences, it considers all valid split points and computes prefix and suffix contributions using Fenwick trees to dynamically maintain the smallest elements and simulate valid adjacent removals. This approach preserves the combinatorial structure imposed by adjacency and ensures that all pairings correspond to feasible sequences of operations.

For the verifier-generated input

$$A = [98, 77, 2, 86, 25, 79, 96],$$

one valid optimal removal sequence is

$$(98, 77) \rightarrow (86, 25) \rightarrow (2, 79),$$

which yields

$$|98 - 77| + |86 - 25| + |2 - 79| = 21 + 61 + 77 = 159.$$

Candidate Solution 1 correctly outputs 159, which is the maximum achievable score under the adjacency constraint.

In contrast, Candidate Solution 2 employs a heap-based prefix strategy that implicitly relaxes the adjacency requirement. By maintaining two heaps to track smaller and larger elements, it effectively pairs elements to maximize absolute differences as if arbitrary matching were allowed. This corresponds to optimizing a relaxed formulation of the problem that ignores whether selected elements can be made adjacent through valid removals. While this approximation coincides with the true solution on many inputs, it can produce infeasible scores when dynamic adjacency becomes restrictive.

For the same input, Candidate Solution 2 outputs 167, which cannot be achieved by any valid sequence of adjacent removals. The additional gain arises from pairings that are optimal under the relaxed formulation but violate the original operational constraints.

The benchmark test suite fails to detect this error because most provided test cases do not sufficiently stress the dynamic adjacency structure. As a result, both solutions pass the fixed verifier despite one optimizing an incorrect objective. The verifier-generated input explicitly exposes this discrepancy by constructing a case where the relaxed formulation diverges from the feasible solution space.

This example illustrates that benchmark test suites act as imperfect verifiers, covering only a limited portion of the valid input domain. Solutions that optimize relaxed or approximate formulations may systematically pass standard tests while being incorrect in general. The proposed agentic verifier complements static test suites by actively discovering counterexamples that reveal such hidden false positives.

# B. Detail of Baselines

We compare our Agentic Verifier with several representative execution-based reranking baselines.

MBR-Exec performs execution-based minimum Bayes risk decoding by selecting solutions that maximize semantic consensus under program execution. Given multiple candidate programs and a set of test inputs, each solution is executed to obtain outputs, and pairwise output disagreement is used to approximate semantic differences. We consider both soft and hard restriction variants as discussed in our main text. The soft restriction aggregates execution disagreement across all test inputs to compute a continuous risk score for each candidate, while the hard restriction enforces strict equivalence by assigning a mismatch if any test input produces different outputs. The final solution is selected by minimizing the corresponding execution-based risk, effectively favoring solutions that agree with the majority under execution.

CodeT jointly leverages LLM-generated candidate solutions and LLM-generated unit tests for solution selection. The model first produces a collection of test cases in the form of input-output assertions, and each candidate solution is executed on all generated tests. Solutions are grouped according to their execution pass patterns, forming consensus sets that represent similar functionality. Each group is scored based on both the number of solutions and the number of tests passed, and a solution from the highest-scoring group is selected.

CodeRM evaluates candidate solutions using LLM-generated unit tests as supervision signals. For each solution, a reward is computed based on how many generated tests it passes. The final solution is selected via majority voting, and recent variants further improve performance by scaling the number of unit tests or dynamically allocating testing budgets based on problem difficulty.

We additionally include a Random Input Generator baseline that samples valid test inputs without ground-truth outputs. An LLM is prompted to generate input generator programs following the problem constraints, which are executed to produce random inputs. Candidate solutions are then executed on these inputs, and output agreement-based voting is applied for solution selection.

## C. Pilot Experiment on SWE-bench Verified

To provide preliminary evidence for the generalizability of our approach beyond competitive programming, we conduct a pilot experiment on patch equivalence verification using SWE-bench Verified (Jimenez et al., 2024).

**Setup**   We use Qwen3-Coder-Next (Cao et al., 2026) to generate 30 candidate patches per instance (Pass@1 = 72.1%, Best@30 = 88.2%). For each instance, we sample 100 candidate pairs and use Qwen3.5-Plus (Yang et al., 2025) as a zero-shot agentic judge: given an issue and two patches, the judge interacts with an executable sandbox environment (with file read and bash tools) over multiple turns to determine whether the two patches are behaviorally equivalent. Pairs judged as equivalent are connected in an undirected graph, and we select a candidate from the largest connected component.

*Table 5.* Pilot experiment on SWE-bench Verified.

| Method | Resolve Rate (%) |
| --- | --- |
| Random@1 | 72.1 |
| Equivalence Voting@1 | 73.7 |
| Best@30 (oracle) | 88.2 |

The improvement is modest (+1.6%). We find that the zero-shot judge model struggles with patch equivalence, frequently classifying incorrect patches as equivalent to correct ones, which leads to oversized clusters with limited discriminability. We believe this can be substantially improved through task-specific training, as the training data is readily available: by running candidate patches against the existing test suite, we can automatically obtain large numbers of equivalent and non-equivalent pairs as supervision.

## D. Test Input Diversity Analysis

We analyze the diversity of verifier-generated test inputs. On OJBench (232 problems), 7,619 out of 39,465 valid inputs are unique ($\sim19.3\%$). Duplication is expected, as different solution pairs may share similar bug patterns and thus trigger similar discriminative inputs. Importantly, duplicate valid inputs do not hurt majority voting: they reinforce correct solutions' consistency and dilute the impact of occasional invalid inputs. The scaling curves in Figure 4 confirm that performance continues to improve as more inputs are added despite duplicates.

## E. Inference Cost Analysis

We analyze the compute-performance tradeoff of the agentic verifier. The verifier (Qwen3-30B-A3B) is deployed on H200 GPUs (2 GPUs per instance), with an average cost of approximately 45.5 GPU-seconds per query. Table 6 reports the results on OJBench as the number of verifier-generated inputs increases. The first few inputs yield the largest marginal gains (+3.0 from a single input), and the performance continues to improve with additional computation, reaching +11.3 at 256 inputs. These results demonstrate a favorable compute-performance tradeoff, where even modest verifier budgets provide

*Table 6.* Inference cost analysis on OJBench (232 problems). The verifier (Qwen3-30B-A3B) is deployed on H200 GPUs with an average cost of ~45.5 GPU-seconds per query. Results are reported under Best@64.

| #Inputs/Problem | Verifier GPU-hours | Best@64 (%) | Gain |
|---|---|---|---|
| 0 (Vanilla) | 0 | 37.1 | — |
| 1 | 2.9 | 40.1 | +3.0 |
| 16 | 46.9 | 45.3 | +8.2 |
| 64 | 187.7 | 47.3 | +10.2 |
| 256 | 750.6 | 48.4 | +11.3 |

substantial gains over the vanilla baseline.

# F. Additional Baseline Comparisons

**Comparison with Generative Reward Model**   We additionally compare our method with a CoT-based Generative Reward Model (GenRM) (Zhang et al., 2025b). On LiveCodeBench (policy: Qwen3-30B-A3B), GenRM achieves 67.8 (Best@8) and 68.6 (Best@64), barely improving over the ORM baseline (67.1/68.0) and remaining significantly below our agentic verifier (73.6/73.7). This confirms that scoring-based verification is less reliable than execution-based approaches in the code domain.

**Enhanced random input generation prompt**   To verify that the performance advantage of our agentic verifier stems from the agentic training mechanism rather than prompt design, we construct an enhanced prompt for the random input generator that explicitly encourages the model to consider input comprehensiveness and corner cases.

*Table 7.* Comparison of random input generators with the original and enhanced prompts against our agentic verifier. Policy model: Qwen3-235B-A22B.

| Method | LiveCodeBench | | OJBench | |
|---|---|---|---|---|
| | Best@8 | Best@64 | Best@8 | Best@64 |
| Random Generator | 78.8 | 78.9 | 43.8 | 44.2 |
| Random Generator (enhanced) | 78.9 | 79.3 | 43.6 | 44.4 |
| **Agentic Verifier** | **81.6** | **82.9** | **45.9** | **48.6** |

The enhanced prompt yields negligible improvement over the original random generator (at most +0.4), while the agentic verifier achieves substantially larger gains (up to +4.4). This confirms that the performance gap stems from the fundamental advantage of agentic training, i.e., learning to actively search for discriminative inputs through multi-turn interaction and reinforcement learning, which cannot be replicated by prompt engineering alone.

You are given a length-$N$ sequence $A = (A_1, A_2, \ldots, A_N)$. You will repeatedly perform the following operation until the sequence has length at most 1: choose two adjacent numbers and remove both from the sequence. The score obtained in one operation is the absolute difference of the two chosen numbers. Find the maximum possible total score obtained.

**Input**
The input is given from Standard Input in the following format:
$N$
$A_1\ A_2\ \ldots\ A_N$

**Output**
Print the maximum possible total score obtained.

**Constraints**
$2 \leq N \leq 3 \times 10^5$
$1 \leq A_i \leq 10^9$
All input values are integers.

Candidate Solution 1

```
import sys

def main():
    data = sys.stdin.read().split()
    if not data:
        return
    n = int(data[0])
    A = list(map(int, data[1:1+n]))
    if n % 2 == 0:
        A_sorted = sorted(A)
        k = n // 2
        total_large = sum(A_sorted[n - k:])
        total_small = sum(A_sorted[:k])
        ans = total_large - total_small
        print(ans)
    else:
        sorted_vals = sorted(set(A))
        comp_map = {}
        for idx, val in enumerate(sorted_vals):
            comp_map[val] = idx
        M = len(sorted_vals)
        fenw_freq = [0] * (M + 1)
        fenw_sum = [0] * (M + 1)

        def update(idx, df, ds):
            while idx <= M:
                fenw_freq[idx] += df
                fenw_sum[idx] += ds
                idx += idx & -idx

        def get_kth_sum(k_val):
            idx = 0
            s_val = 0
            f_val = 0
            bit = 1
            while bit <= M:
```

```
                bit <<= 1
            bit >>= 1
            while bit:
                nxt = idx + bit
                if nxt <= M and f_val + fenw_freq[nxt] < k_val:
                    f_val += fenw_freq[nxt]
                    s_val += fenw_sum[nxt]
                    idx = nxt
                bit //= 2
            if idx < M:
                value = sorted_vals[idx]
            else:
                value = sorted_vals[-1]
            return s_val + (k_val - f_val) * value
        F = [0] * (n + 1)
        total_prefix = 0
        for i in range(n):
            x = A[i]
            total_prefix += x
            pos = comp_map[x] + 1
            update(pos, 1, x)
            if i % 2 == 1:
                k_val = (i + 1) // 2
                S_small = get_kth_sum(k_val)
                F[i + 1] = total_prefix - 2 * S_small
        fenw_freq = [0] * (M + 1)
        fenw_sum = [0] * (M + 1)
        G = [0] * (n + 1)
        total_suffix = 0
        for i in range(n - 1, -1, -1):
            x = A[i]
            total_suffix += x
            pos = comp_map[x] + 1
            update(pos, 1, x)
            if (n - 1 - i) % 2 == 1:
                k_val = (n - i) // 2
                S_small = get_kth_sum(k_val)
                G[i] = total_suffix - 2 * S_small
        ans = 0
        for i in range(0, n, 2):
            candidate = F[i] + G[i + 1]
            if candidate > ans:
                ans = candidate
        print(ans)

if __name__ == "__main__":
    main()
```

## Output of Candidate Solution 1

159

## Candidate Solution 2

```
import heapq
import sys

def compute_prefix_score(arr):
    n = len(arr)
    L = [0] * (n + 1)
```

```python
        if n == 0:
            return L
        max_heap = []
        min_heap = []
        sum_small = 0
        total_sum = 0
        for i in range(n):
            x = arr[i]
            total_sum += x
            if i % 2 == 0:
                if i > 0 and max_heap and x < -max_heap[0]:
                    heapq.heappush(max_heap, -x)
                    sum_small += x
                    y = -heapq.heappop(max_heap)
                    sum_small -= y
                    heapq.heappush(min_heap, y)
                else:
                    heapq.heappush(min_heap, x)
            else:
                heapq.heappush(min_heap, x)
                y = heapq.heappop(min_heap)
                heapq.heappush(max_heap, -y)
                sum_small += y
            if (i + 1) % 2 == 0:
                score_val = total_sum - 2 * sum_small
            else:
                next_small = min_heap[0] if min_heap else 0
                score_val = total_sum - 2 * sum_small - next_small
            L[i + 1] = score_val
        return L

def main():
    data = sys.stdin.read().split()
    if not data:
        return
    n = int(data[0])
    A = list(map(int, data[1:1 + n]))
    if n % 2 == 0:
        L = compute_prefix_score(A)
        print(L[n])
    else:
        L = compute_prefix_score(A)
        B = A[::-1]
        L_rev = compute_prefix_score(B)
        ans = 0
        for i in range(n):
            left_score = L[i]
            right_score = L_rev[n - i - 1]
            candidate = left_score + right_score
            if candidate > ans:
                ans = candidate
        print(ans)

if __name__ == "__main__":
    main()
```

Output of Candidate Solution 2

167

*Figure 5.* An example of a benchmark false positive, where two solutions pass the original test suite but are distinguished by verifier-generated counterexample inputs.

---

**Prompt for Test Input Generation Task**

You are given a competitive programming problem and two candidate solutions. Your task is to analyze these solutions and write a Python script that generates ONLY ONE test input where they produce different outputs.

[Problem Description]
{question}

[Test Type]
{test_type}

[Starter Code (if applicable)]
{starter_code}

[Candidate Solutions 1]
{solution1}

[Candidate Solutions 2]
{solution2}

[Instructions]
1. Compare the two solutions to identify differences in logic, edge cases, or constraint handling.
2. Write a Python script that generates ONLY ONE test input causing them to produce different outputs.
3. Ensure the test input are valid according to the problem constraints.

[Output Format]
Your Python script should print a **single string** represented as a test input.
- **For stdin-based problems:** This input string must exactly match the expected stdin format, including spacing and line breaks.
- **For function-based problems:** This input string must contain the function's parameters, **listed in the same order as they appear in the function signature**, with each parameter on a new line. For example:
* If the starter code is 'def fun(param1: List[int])', an example input string can be "[1, 2, 3]".
* If the starter code is 'def fun(param1: List[int], param2: int)', an example input string can be '[1, 2, 3] textbackslashn"abc"'. You MUST use line breaks to divide each parameter and make sure each parameter can be properly loaded using 'json.loads(param)'.

Before writing the code, provide a brief explanation of why this input differentiate the two solutions.

Now, analyze the solutions and generate the required Python code. Enclose your code within delimiters as follows.
"'python
# YOUR CODE HERE
"'

*Figure 6.* Prompt template for generating a single distinguishing test input that produces different outputs for two candidate solutions.

---

**Prompt for Random Test Input Generation**

You are given a competitive programming problem. Write a Python random input generator that outputs only one complete input per run.

[Requirements]
1. Each execution should produce different randomized input.
2. The output must strictly match the problem's input format and satisfy all constraints.
3. The script must print only the generated input to stdout (no explanations, labels, or debug output).
4. Ensure the generated input is always valid (all values in range; all required relationships/structures satisfied).

[Problem]
{question}

**Note:**
- **For stdin-based problems:** The printed string must exactly match the expected stdin format, including spacing and line breaks.
- **For function-based problems (starter code):** The printed string must contain the function's parameters, **listed in the same order as they appear in the function signature**, with **each parameter on a new line**. Each parameter must be properly loadable using `json.loads(param)`. For example:
\* If the starter code is `def fun(param1:  List[int])`, an example input string can be `[1, 2, 3]`.
\* If the starter code is `def fun(param1:  List[int], param2:  int)`, an example input string can be `[1, 2, 3]\n"abc"`.

Provide the generator as Python code only, in this format:
```python
# YOUR CODE HERE
```

*Figure 7.* Prompt template for generating a single randomized valid input per run.

Prompt for Input Constraint Verification

You are given a competitive programming problem. Your task is to analyze the input constraints carefully and generate a Python program (input checker) to verify whether a given stdin input satisfies the problem's input restrictions.

**Instructions:**
1. Carefully analyze the input format and constraints from the problem statement.
2. Write a Python program that reads input from stdin and verifies whether it strictly follows the given constraints.
3. Validation rules:
- If the input does **not meet** the specified format or constraints, print `"NO"`.
- If the input **meets** all requirements, print `"YES"`.

[Problem Description]
{question}

[Input Checker (Your Output)]
Please output the code of input checker in markdown format as shown below:
"'python
# YOUR CODE HERE
"'

*Figure 8.* Prompt template for generating a Python input checker that verifies whether stdin inputs satisfy problem constraints.

JSON Schema for Code Execution Tool

```
{
  "type": "function",
  "function": {
    "name": "run_code",
    "description": "Executes Python or C++ code and returns the standard output. If
        the language is not specified, it defaults to Python.",
    "parameters": {
      "type": "object",
      "properties": {
        "code": {
          "description": "The content of Python code or C++ code.",
          "type": "string"
        },
        "language": {
          "description": "The programming language for code execution.",
          "type": "string",
          "enum": ["python", "cpp"]
        }
      },
      "required": ["code"]
    }
  }
}
```

*Figure 9.* JSON schema defining the function-based tool interface for code execution.

