# OpenReview forum: "Scaling Agentic Verifier for Competitive Coding"
_ICML.cc/2026/Conference — ICML 2026 regular_

### Official Review · Reviewer_zQk3 · 2026-03-09

**Soundness:** 2
**Presentation:** 2
**Significance:** 3
**Originality:** 2
**Overall Recommendation:** 3
**Confidence:** 3

**Summary:**

The paper proposes the use of an agentic verifier to improve code generation capabilities of
LLMs specifically in the context of competitive coding environments. Existing approaches
typically tackled this problem as a (co-)generation of multiple versions of the code and test data,
then each version is executed on these tests and correct solutions would then be selected using
some agreement measure of their outputs or majority voting. The nature of competitive
programming makes this very challenging, especially when it comes generating interesting test
cases with high discriminative power. To this end, the proposed agentic verifier builds on this
and tries to generate intelligent test inputs that break wrong solutions. They achieve this by
training a feedback loop that takes into account the execution environment and therefore can
reason about program behaviours to guide test input generation.

The evaluation tests the proposed agentic verifier on several competitive programming
benchmarks, including USACO, LiveCodeBench, OJBench, ICPC-Eval, and a curated
CodeForces dataset. The system generates multiple candidate solutions using Qwen models
and uses the verifier to produce discriminative test inputs that help rerank these solutions. Its
performance is compared against several baselines such as reward-model ranking,
execution-based voting methods, and random input generation. Results show that the agentic
verifier consistently achieves the highest accuracy under both Best@8 and Best@64 evaluation
settings. Additional analyses indicate that the method scales well with increased test-time
computation and provides larger improvements on more difficult programming problems

**Compliance With Llm Reviewing Policy:**

Affirmed.

**Key Questions For Authors:**

- **Q1** The authors also introduce their Execution-based Voting method in Section 2.1. What
was the main reason behind this custom voting mechanism? Do existing methods fail on
these competitive programming tasks? I believe there exists a large body of work on
voting mechanisms in the literature, e.g., the most recent and related would be [1]

- **Q2**  Would established LLM-powered fuzzing techniques also struggle in finding these
discriminative inputs? The idea behind fuzzing would then provide useful seed inputs
(preferably those that show this behaviour of interest) and through mutations try to
discover more test inputs.

- **Q3** Since the evaluation is based on counting the number of test inputs in Figures 1 and 4. I
find this not that insightful. How diverse (or similar) are these generated discriminative
test inputs? I find using count as not helpful because even with different generators,
there is a non-zero chance of generating duplicates. This is not (explicitly) clear from
reading the paper. Did the authors de-duplicate these inputs?

### References:

[1]: Thomas Valentin, Ardi Madadi, Gaetano Sapia, Marcel Böhme: Estimating Correctness
Without Oracles in LLM-Based Code Generation.

**Limitations:**

Yes

**Strengths And Weaknesses:**

## Strengths
- Well motivated
- A well engineered and scalable solution

## Weakness
- Novelty feels incremental
- Some technical details are missing in order to properly judge the soundness of the
approach

## Originality (fair)

The conceptual contribution of the paper feels incremental at best: from a software testing
perspective this feels like another LLM-powered fuzzing technique of which several tools and
techniques now exist. The generation of many solutions and selecting the best is not
necessarily new. However, the technical solution is neat because it combines and builds on
these different techniques into a scalable and well engineered solution especially given the
challenges faced by current solutions in the context of competitive programming.

## Soundness (poor/fair)

While the approach is shown to outperform a diverse set of baselines, in fact, it significantly
beats all of these baselines in the Best@8 and Best@64 metrics, I have some concerns
especially on the design of the training pipeline in Section 3. The training pipeline relies on
crawled competitive programming problems and submissions, but the paper does not clearly
describe how overlap with evaluation benchmarks is prevented. Since the verifier is trained on
problem solution pairs from similar sources (e.g., online judges), it is possible that benchmark
problems or closely related submissions appear in the training data. This raises concerns about
potential data contamination and makes it difficult to determine whether the reported
improvements reflect true generalization.

## Presentation (fair)
The paper is by and large easy to follow. It is well written and I really appreciate the experiment
based motivation shown by the plots in Figure 1. However, there are several places where I
believe it can be improved in order for readers to understand. I did not quite understand this
phenomenon of imperfect benchmark verifiers in the introduction. See questions and minor
comments for further areas where I feel like the presentation can be improved.

## Significance (good)

The paper tackles one of the main challenges of LLM generated code - trustworthiness of the
generated artifacts. It builds on the established co-generation of code (multiple versions of) and
tests and then uses voting mechanisms based on execution outcome to select the best version.
Obviously the authors demonstrate that this established framework is not directly transferable in
competitive coding environments and propose a feedback directed approach to tackle this
problem. Therefore the paper solves a relevant problem.

---

> ### Author Rebuttal · Authors · 2026-03-31
>
> We sincerely thank the reviewer for the detailed and thoughtful evaluation. We are encouraged that the reviewer finds our work well-motivated, solving a relevant problem in LLM code trustworthiness, and appreciates the experiment-based motivation (Figure 1) as well as the scalable and well-engineered technical solution. We address each concern below.
>
> ---
>
> > **Soundness**: "How is overlap with evaluation benchmarks prevented?"
>
> Thanks for raising this important point. We perform explicit deduplication between the training data and all five evaluation benchmarks. Specifically, we compute pairwise problem similarity using **Qwen3-Embedding-8B** (SOTA on MTEB Eng v2), and remove all training problems with similarity > 0.7 to any evaluation problem. This ensures no data leakage at the problem level. We apologize for not including this detail and will add it in the revised paper.
>
> ---
>
> > **Presentation**: "I did not quite understand imperfect benchmark verifiers."
>
> Thanks for the feedback. In competitive programming, correctness is determined by a **fixed set of test cases** covering only a limited portion of the valid input space. Solutions passing all tests are not necessarily correct on all valid inputs and such solutions are **false positives**. As shown in Section 5.4 (Figure 5), two solutions both labeled correct produce different outputs (159 vs. 167) on a verifier-generated input, revealing one violates the problem's dynamic adjacency constraint.
>
> Our verifier can actively expose such false positives by generating targeted discriminative inputs, serving as a complementary verification tool beyond best-of-N selection. We agree that this concept was not introduced clearly enough and will improve the presentation in the revised paper.
>
> ---
>
> > **Q1**: "What was the main reason behind this custom voting mechanism? Do existing methods fail on these competitive programming tasks?"
>
> Thanks for the question. Our voting mechanism is based on a simple and intuitive principle: on a valid test input, **correct solutions always produce consistent outputs**, while correct-incorrect pairs and incorrect-incorrect pairs may produce different outputs. The voting strategy (Section 2.1) directly operationalizes this idea by clustering solutions by output agreement and selecting the majority cluster.
>
> We note that output-agreement-based selection mechanisms are common in the literature (e.g., AlphaCode [1]). The referenced work [2] shares a similar intuition, using behavioral disagreement (incoherence) between programs as a proxy for correctness estimation, and provides a theoretical guarantee that incoherence lower-bounds error. Our work differs in two aspects: (1) [2] focuses on **estimating** program correctness, while we focus on **selecting** the best candidate via voting; (2) [2] uses fuzzing-generated inputs, while our key contribution is training an agentic verifier to generate **discriminative** inputs that maximize behavioral divergence, which as discussed in Q2 is difficult for fuzzing in competitive programming. We appreciate the reference and will discuss it in the revised paper.
>
> ---
>
> > **Q2**: "Would LLM-powered fuzzing techniques struggle in finding discriminative inputs?"
>
> Thanks for the interesting question. Fuzzing relies on good seed inputs and mutation strategies to explore the input space. However, in competitive programming, the only available seeds are the few trivial examples in the problem statement (typically 1–3 cases). Without diverse and representative seeds, fuzzing degenerates into **random input generation**, which we show to be inefficient (Section 2, Figure 1). Our verifier addresses this limitation: it does not rely on seed inputs and is trained via RL to directly optimize for discriminative power between candidate solutions.
>
> ---
>
> > **Q3**: "How diverse are the generated test inputs? Did the authors de-duplicate?"
>
> Thanks for the question. We do not deduplicate. The verifier's objective is generating **discriminative** test inputs (i.e., distinguishing a given solution pair), rather than targeting diversity.
>
> On OJBench (232 problems), 7,619 out of 39,465 valid inputs are unique (~19.3%). Duplication is expected as different solution pairs may share similar bug patterns. Duplicate valid inputs do not hurt majority voting. They reinforce correct solutions' consistency and dilute the impact of occasional invalid inputs. The scaling curves (Figure 4) confirm performance continues to improve as more inputs are added despite duplicates. We will add this analysis in the revised paper.
>
> ---
>
> *reference*:
>
> [1] Li Y, Choi D, Chung J, et al. Competition-level code generation with alphacode. Science, 2022.
>
> [2] Thomas Valentin, Ardi Madadi, Gaetano Sapia, Marcel Böhme: Estimating Correctness Without Oracles in LLM-Based Code Generation.

---

> > ### Author Rebuttal · Reviewer_zQk3 · 2026-04-05
> >
> > My question and concerns have been adequately addressed.

---

> > > ### Author Response · Authors · 2026-04-05
> > >
> > > We appreciate the reviewer's careful evaluation and are glad our response fully addressed the concerns. We will incorporate the additional analyses into the revised paper. We would also be grateful if the reviewer could reconsider the overall assessment in light of the rebuttal.

---

### Official Review · Reviewer_eRXz · 2026-03-10

**Soundness:** 3
**Presentation:** 3
**Significance:** 3
**Originality:** 3
**Overall Recommendation:** 5
**Confidence:** 3

**Summary:**

This paper proposes an Agentic Verifier for competitive programming, aiming to improve execution-based reranking by actively generating highly discriminative test inputs. Instead of relying on randomly generated inputs or full test-case synthesis, the verifier interacts with a code execution environment in a multi-turn manner to search for inputs that expose behavioral discrepancies between candidate solutions. The model is trained through a scalable pipeline combining large-scale data synthesis, rejection fine-tuning on successful interaction trajectories, and agentic reinforcement learning.
Extensive experiments across multiple competitive programming benchmarks show that the proposed verifier consistently improves Best-of-N selection performance over strong execution-based baselines and zero-shot alternatives. The method demonstrates clear test-time scaling behavior and is particularly effective on harder problems, suggesting that learned, execution-guided input generation can significantly enhance reranking quality.

**Compliance With Llm Reviewing Policy:**

Affirmed.

**Final Justification:**

I believe the author's current rebuttal has largely addressed my concerns. In my opinion, the current paper meets the standard for acceptance.

**Key Questions For Authors:**

### Key Questions for Authors

1. **Strength of the random input baseline.**
   The current random input generator baseline does not appear to explicitly encourage adversarial or corner-case-oriented test construction. Could the authors clarify whether stronger prompting strategies were explored for the zero-shot baseline (e.g., explicitly instructing the model to search for edge cases or adversarial inputs)? Providing results with such stronger baselines would help determine whether the gains stem primarily from the agentic interaction mechanism rather than prompt design.

2. **Training–evaluation dataset separation.**
   Could the authors clarify whether the training data used for the agentic verifier is strictly disjoint from the evaluation benchmarks (e.g., USACO, LiveCodeBench, OJBench, ICPC-Eval)? In particular, are there any overlaps at the problem level, or similar tasks that might allow the verifier to learn benchmark-specific patterns? A clear statement about dataset separation would help assess the reliability of the reported generalization results.

3. **Potential use of the verifier for improving the base policy model.**
   The proposed agentic verifier appears to generate rich supervision signals through adversarial test inputs and execution feedback. Have the authors explored whether these signals could be leveraged to improve the underlying policy model itself (e.g., through reinforcement learning or additional fine-tuning), rather than only for test-time reranking? Discussion or preliminary experiments in this direction could broaden the impact of the method.

**Limitations:**

yes

**Strengths And Weaknesses:**

### Strengths

- **Well-motivated and carefully designed methodology.**
  The paper clearly identifies the inefficiency of naive random input scaling in execution-based reranking and proposes a principled agentic verifier that actively searches for discriminative test inputs through multi-turn interaction. The overall training pipeline (data synthesis → rejection fine-tuning → reinforcement learning) is thoughtfully constructed and technically sound.

- **Strong and consistent empirical improvements.**
  Across multiple competitive programming benchmarks and two policy models, the proposed method consistently outperforms strong execution-based baselines. The gains are substantial in absolute terms and particularly pronounced on harder benchmarks.

- **Comprehensive experimental evaluation.**
  The authors evaluate across multiple datasets, model scales, and Best@k settings, and further analyze scaling behavior and problem difficulty. The inclusion of zero-shot baselines helps demonstrate that the improvements are due to agentic training rather than model size alone.

- **Clear demonstration of test-time scaling benefits.**
  The method shows stable performance improvements as the number of candidate solutions and generated inputs increases, indicating that it effectively leverages additional computation.

### Weaknesses

1. **Strength of the random input baseline.**
   It is unclear whether the random input generator baseline is sufficiently strong. The prompt does not appear to explicitly encourage adversarial or corner-case-oriented input construction. A stronger zero-shot baseline with explicit corner-case reasoning could help clarify whether the observed gap is due to the lack of agentic interaction or simply prompt design.

2. **Dataset separation between training and evaluation.**
   It is unclear whether the training data used for the agentic verifier is strictly disjoint from the evaluation benchmarks (e.g., USACO, LiveCodeBench, OJBench, ICPC-Eval). If there is overlap at the problem level, the verifier may implicitly learn problem-specific structures or common failure modes, which could inflate evaluation performance. The paper would benefit from a clear statement regarding dataset separation.

3. **Potential for improving the base policy model.**
   While the paper demonstrates that the agentic verifier significantly improves test-time reranking performance, it remains unclear whether the verifier could also provide useful supervision signals to further improve the underlying policy model itself. The current work focuses solely on test-time selection, and it would strengthen the contribution to explore whether the verifier can be integrated into a reinforcement learning framework to enhance the base model.

---

> ### Author Rebuttal · Authors · 2026-03-31
>
> We sincerely thank the reviewer for the thorough evaluation. We are encouraged that the reviewer finds our methodology well-motivated and carefully designed, the training pipeline technically sound, the empirical improvements strong and consistent (particularly on harder benchmarks), and the test-time scaling behavior clearly demonstrated. We address each question below.
>
> ---
>
> > Weakness 1 & Key Question 1: "It is unclear whether the random input generator baseline is sufficiently strong. The prompt does not appear to explicitly encourage adversarial or corner-case-oriented input construction."
>
> Thanks for the suggestion. Our original prompt for the random input generator primarily focuses on input validity (i.e., satisfying problem constraints). To address this concern, we construct an **enhanced prompt** that additionally encourages the model to consider input **comprehensiveness** and **corner cases**, and conduct a supplementary experiment.
>
> Results on LiveCodeBench and OJBench (policy model: Qwen3-235B-A22B, Best@8 / Best@64):
>
> | Method | LCB Best@8 | LCB Best@64 | OJB Best@8 | OJB Best@64 |
> |:--|:-:|:-:|:-:|:-:|
> | Random Generator | 78.8 | 78.9 | 43.8 | 44.2 |
> | Random Generator (enhanced prompt) | 78.9 (+0.1) | 79.3 (+0.4) | 43.6 (-0.2) | 44.4 (+0.2) |
> | Agentic Verifier | 81.6 (+2.8) | 82.9 (+4.0) | 45.9 (+2.1) | 48.6 (+4.4) |
>
> The enhanced prompt yields negligible improvement over the original random generator (at most +0.4), while the Agentic Verifier achieves substantially larger gains (up to +4.4). This suggests that the performance gap is not due to prompt design, but rather stems from the fundamental advantage of **agentic training**, learning to actively search for discriminative inputs through multi-turn interaction and RL, which cannot be replicated by prompt engineering alone.
>
> ---
>
> > Weakness 2 & Key Question 2: "Could the authors clarify whether the training data used for the agentic verifier is strictly disjoint from the evaluation benchmarks?"
>
> Thanks for raising this important point. We perform explicit deduplication between the training data and all five evaluation benchmarks. Specifically, we compute pairwise problem similarity using **Qwen3-Embedding-8B**, which achieves SOTA performance on MTEB (Eng v2), and remove all training problems with similarity > 0.7 to any evaluation problem. This ensures there is no data leakage at the problem level.
>
> We appreciate the reviewer for highlighting this. We will add a clear description of this deduplication procedure in the revised paper.
>
> ---
>
> > Weakness 3 & Key Question 3: "Have the authors explored whether these signals could be leveraged to improve the underlying policy model itself?"
>
> Thanks for the insightful suggestion. While we have not conducted experiments on directly improving the policy model in this work, we believe our agentic verifier provides a natural foundation for this direction through two complementary paths:
>
> 1. **Enhancing existing test cases for more precise rewards.** As discussed in Section 5.4, fixed benchmark test suites often suffer from false positives (incorrect solutions that pass all tests). Our verifier can augment existing datasets by generating additional discriminative test inputs, reducing both false positives and false negatives. This yields more precise reward signals for policy model training via RFT and RL [1, 2].
>
> 2. **Constructing test cases for new data to enable data scaling.** For problems that have labeled solutions but lack test cases, our verifier can construct discriminative test case sets from scratch. This unlocks additional training data for policy model improvement, enabling data scaling for RFT and RL.
>
> We will include this discussion in the revised paper.
>
> ---
>
> *reference*:
>
> [1] Wang Z, Liu S, Sun Y, et al. CodeContests+: High-Quality Test Case Generation for Competitive Programming[J].
>
> [2] Zhou S, Zheng Z, Liu K, et al. AutoCode: LLMs as Problem Setters for Competitive Programming. arXiv preprint arXiv:2510.12803, 2025.

---

> > ### Author Rebuttal · Reviewer_eRXz · 2026-04-02
> >
> > I believe the author's current rebuttal and additional experiments has largely addressed my concerns.

---

> > > ### Author Response · Authors · 2026-04-05
> > >
> > > We appreciate the reviewer's careful evaluation and are glad our rebuttal and additional experiments addressed the concerns. We will incorporate these results into the revised paper.

---

### Official Review · Reviewer_G4LN · 2026-03-13

**Soundness:** 3
**Presentation:** 3
**Significance:** 3
**Originality:** 2
**Overall Recommendation:** 4
**Confidence:** 4

**Summary:**

This paper addresses the verification problem in code generation, specifically selecting the correct candidate from multiple generated programs by comparing their execution outputs. Existing methods generate random inputs and compare execution results across candidates, but most random inputs fail to expose differences between candidates, leading to low verification efficiency. To address this, the authors train an agent to actively search for discriminative inputs that produce differing outputs across candidate code pairs, using a multi-turn interaction framework. The training pipeline combines data synthesis, rejection fine-tuning (SFT), and reinforcement learning. Consistent performance improvements over random input-based verification are reported across multiple coding benchmarks.

**Compliance With Llm Reviewing Policy:**

Affirmed.

**Final Justification:**

The authors have adequately addressed most of my concerns with additional experiments and clear explanations, particularly the training ablation and inference cost analysis. I am raising my score, assuming these results are incorporated into the revised paper.

**Key Questions For Authors:**

**Q1**. What is the additional inference FLOPs or wall-clock cost of the verifier agent relative to its performance gains, and is there an analysis of the compute-performance trade-off?

**Q2**. Can comparisons against other verifier or reward model-based approaches such as GenRM [1], ORM [2], or PRM be provided?

**Q3**. Are there ablation results for the training pipeline comparing SFT-only, RL-only, and SFT+RL configurations?

[1] Zhang et al., Generative Verifiers: Reward Modeling as Next-Token Prediction, ICLR, 2024

[2] Luo et al., Improve Mathematical Reasoning in Language Models by Automated Process Supervision, arXiv, 2024

**Limitations:**

The discussion on limitations is somewhat limited.

**Strengths And Weaknesses:**

### Strengths

- Reframing verification as discriminative input search rather than random input sampling is an interesting and well-motivated reformulation.
- The multi-turn interaction structure for iteratively finding distinguishing inputs is intuitive and convincing.
- Consistent improvements over random input verification baselines are demonstrated across multiple coding benchmarks.

---

### Weaknesses
1. Mismatch between pairwise training and multi-candidate inference.
- The verifier is trained via pairwise code comparison, but inference requires selecting one correct candidate from multiple options. Whether pairwise comparisons reliably aggregate into a globally correct selection is not analyzed, and this setting mismatch is not adequately addressed.
2. Disagreement does not guarantee correctness.
- The reward signal is based on finding inputs that produce differing outputs between two candidate programs. However, the paper does not sufficiently discuss whether disagreement reliably indicates correctness. When both candidates produce incorrect outputs, the verifier may still make an incorrect selection, and this failure mode is not analyzed.
3. Limited inference cost analysis.
- Despite requiring multi-turn interactions and repeated code executions, the paper provides only limited inference cost analysis. Comparisons based on number of inputs alone are insufficient; additional analysis of verifier inference FLOPs and wall-clock time is needed to fairly assess the practical computational overhead.
4. Narrow baseline comparisons.
- The paper compares primarily against random input verification. Including comparisons with other verifier and reward model-based approaches, such as GenRM, ORM, and PRM, would more clearly situate the proposed method and demonstrate its relative advantages.
5. Insufficient ablation of the training pipeline.
- The training pipeline consists of data synthesis, SFT, and RL, but ablation studies examining the contribution of each stage are lacking. In particular, a comparison between SFT-only and SFT+RL would clarify the necessity and added value of the reinforcement learning stage.

---

> ### Author Rebuttal · Authors · 2026-03-31
>
> We sincerely thank the reviewer for the thorough feedback. We are glad that the reviewer finds our reformulation interesting and well-motivated, the multi-turn structure intuitive, and the improvements consistent. We address each concern below.
>
> ---
>
> > **W1**: "Mismatch between pairwise training and multi-candidate inference."
>
> Thanks for your question. The pairwise formulation is connected to multi-candidate inference through the voting mechanism (Section 2.1).
>
> 1. **Pairwise as atomic unit.** Distinguishing two solutions is a binary classification problem, whereas distinguishing among multiple candidates becomes a clustering problem that is harder to optimize. Pairwise comparison provides a simple yet effective paradigm.
>
> 2. **Aggregation at inference.** At inference, we sample candidate pairs, generate one discriminative input per pair, then execute **all** N candidates on **all** inputs. The voting strategy aggregates pairwise-originated inputs into a global ranking via output agreement.
>
> 3. **Empirical validation.** Table 1 confirms this pipeline consistently improves selection across all five benchmarks and two policy models.
>
> We will add this design rationale in the revised paper.
>
> ---
>
> > **W2**: "Disagreement does not guarantee correctness."
>
> Thanks for your question. In our framework, disagreement is **not directly used to judge correctness**. It is a training signal for generating discriminative inputs; correctness is determined by **majority voting** (Section 2.1).
>
> 1. **Why voting works.** On discriminative inputs, correct solutions cluster together (same correct output), while incorrect ones scatter. This makes the correct cluster more prominent.
>
> 2. **Wrong-wrong disagreement is beneficial.** Incorrect solutions disagreeing splits their votes, making it harder for them to form a false majority.
>
> 3. **Robustness through aggregation.** Many inputs are generated and all N candidates run on all of them. Voting over many inputs ensures the correct cluster receives the most votes overall.
>
> We will include this discussion in the revised paper.
>
> ---
>
> > **W3 & Q1**: "Limited inference cost analysis."
>
> Thanks for your question. We deploy the verifier (Qwen3-30B-A3B) on H200 GPUs (2 GPUs/instance). From 7,168 sampled queries on 32 GPUs (2h50min), the average cost is **~45.5 GPU-seconds per query**.
>
> Compute-performance tradeoff on OJBench (232 problems, Best@64):
>
> | #Inputs/Problem | Verifier GPU-hours | Best@64 (%) | Gain |
> |:-:|:-:|:-:|:-:|
> | 0 (Vanilla) | 0 | 37.1 | — |
> | 1 | 2.9 | 40.1 | +3.0 |
> | 16 | 46.9 | 45.3 | +8.2 |
> | 64 | 187.7 | 47.3 | +10.2 |
> | 256 | 750.6 | 48.4 | +11.3 |
>
> The first few inputs yield the largest marginal gains. We will include this analysis in the revised paper.
>
> ---
>
> > **W4 & Q2**: "Narrow baseline comparisons. Can GenRM, ORM, or PRM comparisons be provided?"
>
> Thanks for your question. Our baselines already cover multiple verification paradigms: **ORM** (Grading RM with Skywork-Reward-V2), **full test case generation** (CodeT, CodeRM), and **input-only generation** (MBR-Exec, Random Generator).
>
> We additionally run a **CoT-based GenRM** on LiveCodeBench (policy: Qwen3-30B-A3B):
>
> | Method | Best@8 | Best@64 |
> |:--|:-:|:-:|
> | Grading RM (ORM) | 67.1 | 68.0 |
> | GenRM (Qwen3-30B-A3B) | 67.8 | 68.6 |
> | Random Generator | 70.7 | 70.4 |
> | **Agentic Verifier** | **73.6** | **73.7** |
>
> GenRM barely improves over ORM and remains far below execution-based methods, confirming that scoring-based verification is less reliable than execution-based approaches in the code domain. Regarding PRM, to our knowledge there is no effective PRM for competitive coding yet.
>
> We will include the GenRM experiment and a clearer baseline taxonomy in the revised paper.
>
> ---
>
> > **W5 & Q3**: "Insufficient ablation of training pipeline (SFT-only, RL-only, SFT+RL)."
>
> Thanks for your question. Regarding **RL-only**: training starts from a pretrained base model lacking instruction-following capabilities, so RL alone cannot produce meaningful rollouts. We therefore do not include RL-only.
>
> Ablation following Table 3 setting (policy: Qwen3-235B-A22B, 64 test inputs):
>
> | Method | LiveCodeBench | OJBench |
> |:--|:-:|:-:|
> | Zero-shot (Qwen3-30B-A3B) | 78.8 (+4.1) | 43.3 (+6.2) |
> | RFT-only (Qwen3-30B-A3B) | 79.9 (+5.2) | 44.6 (+7.5) |
> | RFT + RL (Qwen3-30B-A3B) | 82.4 (+7.7) | 47.3 (+10.2) |
>
> RFT-only improves marginally over zero-shot because it generates a high rate of **invalid inputs** (Figure 3 middle). RL explicitly penalizes invalid outputs (reward = -1), substantially reducing the invalid rate and increasing discriminative power.
>
> We will include this ablation in the revised paper.
>
> ---
>
> *reference*:
>
> [1] Zhang et al., Generative Verifiers: Reward Modeling as Next-Token Prediction, ICLR, 2024

---

> > ### Author Rebuttal · Reviewer_G4LN · 2026-04-02
> >
> > Thanks for the thorough response. The authors have adequately addressed most of my concerns with additional experiments and clear explanations, particularly the training ablation and inference cost analysis. I am raising my score, assuming these results are incorporated into the revised paper.

---

> > > ### Author Response · Authors · 2026-04-05
> > >
> > > We appreciate the reviewer's thorough evaluation and are glad our response addressed the concerns. We will incorporate all supplementary experiments and analyses into the revised paper.

---

### Official Review · Reviewer_VhpU · 2026-03-23

**Soundness:** 2
**Presentation:** 2
**Significance:** 2
**Originality:** 2
**Overall Recommendation:** 4
**Confidence:** 4

**Summary:**

This paper proposes an agentic workflow for generating differentiating test cases for competitive programming. The idea is based on evaluating generated test cases by their ability to differentiate solutions whose correctness is labeled with ground-truth tests. Based on the workflow, it collects training data (problem, solutions with different labels) and applies rejection fine-tuning and reinforcement learning to improve the underlying model. It is shown to be effective at reranking llm-generated solutions to select the best ones.

**Compliance With Llm Reviewing Policy:**

Affirmed.

**Final Justification:**

The author's rebuttal has addressed my concerns about the contribution of the method.

**Key Questions For Authors:**

1. What aspects of the pipeline can go beyond competitive programming?

2. Why should people continue the research on generating verifiers for competitive programming when LLMs are already doing so well and winning IOI gold medals?

**Limitations:**

Yes.

**Strengths And Weaknesses:**

Strengths:
1. It combines both agentic workflows and workflow-targetd LLM post training to achieve good results with a small model.
2. The method clearly works compared to the baselines listed.

Weakness:
1. The proposed agentic workflow is very similar to those proposed in [1 - 4], and the training procedure is similar to what people do for all types of LLM-post training. Very limited technical contribution to the field.
2. Limited discussion about how this pipeline can work beyond the domain of competitive programming.


[1] CodeContests+: High-Quality Test Case Generation for Competitive Programming, https://arxiv.org/abs/2506.05817
[2] HardTests: Synthesizing High-Quality Test Cases for LLM Coding, https://arxiv.org/abs/2505.24098
[3] AutoCode: LLMs as Problem Setters for Competitive Programming, https://arxiv.org/abs/2510.12803
[4] Klear-CodeTest: Scalable Test Case Generation for Code Reinforcement Learning, https://arxiv.org/abs/2508.05710

---

> ### Author Rebuttal · Authors · 2026-03-31
>
> We thank the reviewer for the evaluation. We are glad that the reviewer recognizes the effectiveness of combining agentic workflows with targeted LLM post-training, and that the method clearly works compared to baselines. We address each concern below.
>
> ---
>
> > **W1**: "The proposed agentic workflow is very similar to [1-4]. Very limited technical contribution."
>
> Thanks for the comment. We respectfully argue that our approach is **fundamentally different** from the four cited works in both methodology and objective.
>
> 1. **Methodological difference.** [1-4] all adopt a **Generator-Validator pipeline**, where both the test input generator and the input validator are prompt-based LLMs. In contrast, our method formulates discriminative test input generation as a **trainable, multi-turn agentic task**. Through rejection fine-tuning and agentic RL with rule-based outcome reward, the capabilities of both generation and validation are **internalized into a single trained verifier**. As shown in Figure 3, the model directly produces inputs that are both **valid** and **discriminative**, without relying on a separate prompt-based validator. This training-based paradigm is scalable, requires no hand-crafted prompt engineering, and is a clear technical departure from the prompt-based pipeline of [1-4].
>
> 2. **Technical contribution.** Our work draws on the recent RLVR (RL with Verifiable Rewards) paradigm and, to our knowledge, is among the first to apply it to test input generation in an agentic setting. The key insight, formulating *distinguishing two candidate solutions as a verifiable reward signal*, enables scalable RL training without human annotation. Our trained 30B verifier outperforms the zero-shot 235B model (Table 3), demonstrating that this formulation is both effective and non-trivial.
>
> We acknowledge that our paper may not have articulated these distinctions clearly enough. We will incorporate [1-4] into the related work with an explicit comparison.
>
> ---
>
> > **W2 & Q1**: "Limited discussion about how this pipeline can work beyond competitive programming." / "What aspects can go beyond competitive programming?"
>
> Thanks for the suggestion. Our pipeline is indeed designed for competitive programming, motivated by the observation that generating complete test cases is often as difficult as solving the problem itself (Section 1). However, the underlying **task formulation**, given a problem and two candidate answers, generating tests to distinguish them, has broader applicability.
>
> Behavioral equivalence testing is a fundamental need across many code domains [5]. For instance, one could train an agentic verifier to generate repo-level unit tests that distinguish between two candidate patches, improving verification for automated program repair and code editing. More broadly, improving a model's **verification ability** can generalize to enhance its overall code capabilities, as verification is widely demanded in real-world software development. We will expand this discussion in the revised paper.
>
> ---
>
> > **Q2**: "Why should people continue research on verifiers when LLMs are already winning IOI gold medals?"
>
> Thanks for the question. We argue that the current success of LLMs in competitive programming actually **strengthens** the case for investing in verification research, for the following reasons:
>
> 1. **Generation-verification imbalance.** While LLMs have achieved impressive problem-solving performance, their verification ability has not kept pace. Winning an IOI gold medal places a model at the level of strong human contestants, but it does not mean the model can reliably *judge* whether a given solution is correct. Our work directly addresses this imbalance by improving the model's ability to verify candidate solutions through discriminative test input generation.
>
> 2. **Verification is essential for continued scaling.** To move beyond gold-medal-level performance toward surpassing the strongest human competitors, we believe both generation and verification capabilities must improve *in tandem*. As demonstrated by recent work such as DeepSeekMath-V2 [6], iteratively improving generators and verifiers is a promising paradigm for sustained capability gains. Our agentic verifier contributes to this direction by providing a stronger verification signal that can enable better solution selection and potentially guide future generator training.
>
> In short, strong problem-solving ability makes strong verification *more* important, not less. It is the bottleneck that must be addressed to unlock further scaling.
>
> ---
>
> *reference*:
>
> [5] Disproving program equivalence with llms, https://arxiv.org/abs/2502.18473
>
> [6] Deepseekmath-v2: Towards self-verifiable mathematical reasoning, https://arxiv.org/abs/2511.22570

---

> > ### Author Rebuttal · Reviewer_VhpU · 2026-04-02
> >
> > I'm not fully convinced that this can generalize beyond competitive programming but would love to see evidence if any.

---

> > > ### Author Response · Authors · 2026-04-05
> > >
> > > We sincerely thank the reviewer for the constructive feedback. The concern about generalization is well-taken and is something we are actively thinking about.
> > >
> > > A natural direction for generalization is **program equivalence verification**, which has been studied across diverse domains such as mutant detection [1], SQL equivalence [2], and patch equivalence [3]. The core task, determining whether two programs are behaviorally equivalent, shares the same structure as our competitive programming setting: given two candidates, use execution-based testing to distinguish their behavior. To provide initial evidence, we conducted a pilot experiment on patch equivalence using SWE-bench Verified.
> > >
> > > **Setup.** We use Qwen3-Coder-Next to generate 30 candidate patches per instance (Pass@1 = 72.1%, Pass@30 = 88.2%). For each instance, we sample 100 candidate pairs and use Qwen3.5-Plus as an agentic judge: given an issue and two patches, the judge interacts with an executable sandbox environment (with file read and bash tools) over multiple turns to determine whether the two patches are behaviorally equivalent. Pairs judged as equivalent are connected in an undirected graph, and we select a candidate from the largest connected component.
> > >
> > > | Method | Resolve Rate (%) |
> > > |:--|:-:|
> > > | Pass@1 | 72.1 |
> > > | Equivalence Voting@1 | 73.7 (+1.6) |
> > > | Pass@30 (oracle) | 88.2 |
> > >
> > > The improvement is modest (+1.6%). We find that the zero-shot judge model struggles with patch equivalence, frequently classifying incorrect patches as equivalent to correct ones, which leads to oversized clusters with limited discriminability. We believe this can be substantially improved through task-specific training, and the training data is readily available: by running candidate patches against the existing test suite, we can automatically obtain large numbers of equivalent and non-equivalent pairs as supervision.
> > >
> > > Thanks again for your insightful questions. We plan to investigate this direction in future work.
> > >
> > > *reference*:
> > >
> > > [1] Large Language Models for Equivalent Mutant Detection: How Far Are We? https://arxiv.org/abs/2408.01760
> > >
> > > [2] Exploring the Use of LLMs for SQL Equivalence Checking, https://arxiv.org/abs/2412.05561
> > >
> > > [3] Agentic Code Reasoning, https://arxiv.org/pdf/2603.01896v2

---

### Decision · Program_Chairs · 2026-04-30

**Decision:**

Accept (regular)

**Comment:**

This paper proposes an Agentic Verifier for execution-based reranking in competitive programming. Rather than relying on random input sampling or full test case synthesis, the system trains an LLM agent via large-scale data synthesis, rejection fine-tuning (RFT), and agentic reinforcement learning to actively generate discriminative test inputs that expose behavioral discrepancies among candidate solutions. The method shows consistent and substantial improvements across various competitive programming benchmarks.

The reviewers agree that the reformulation of verification as discriminative input search via multi-turn agentic interaction is well-motivated and sound. The training pipeline from data synthesis to RFT and then to agentic RL is well-designed and ablated in the rebuttal, demonstrating the necessity of each stage. In particular, RL substantially reduces invalid input rates beyond what RFT alone achieves. The metareviewer acknowledges that empirical results are strong and consistent and the rebuttal provided high-quality additional experiments.

In the revision, the authors should explicitly state the key difference from concurrent work on discriminative test case generation, particularly the use of a single end-to-end agent trained via RLVR, as opposed to prompt-based Generator–Validator pipelines. Generalization beyond competitive programming remains an open question. This should be clearly framed as future work, along with a discussion of the pilot experiment on SWE-bench Patch. The current manuscript also lacks important details, such as ablation results and inference cost analysis, which were only presented in the rebuttal. These should be incorporated into the final version.

Overall, reviewer concerns were all addressed during the rebuttal period, and the three reviewers who updated their assessments all support acceptance.